# Unpacking Phthalates from Obscurity in the Environment

**DOI:** 10.3390/molecules29010106

**Published:** 2023-12-23

**Authors:** Marzieh Baneshi, Jamey Tonney-Gagne, Fatima Halilu, Kavya Pilavangan, Ben Sabu Abraham, Ava Prosser, Nikaran Kanchanadevi Marimuthu, Rajendran Kaliaperumal, Allen J. Britten, Martin Mkandawire

**Affiliations:** 1Department of Chemistry, School of Science and Technology, Cape Breton University, 1250 Grand Lake Road, Sydney, NS B1P 6L2, Canadacbu22bxtf@cbu.ca (F.H.); cbu17gpz@cbu.ca (K.P.); cbu17fnz@cbu.ca (B.S.A.); cbu16gcf@cbu.ca (A.P.); nikaran_km@cbu.ca (N.K.M.); raj_kalia@cbu.ca (R.K.); allen_britten@cbu.ca (A.J.B.); 2Engineering Co-op Intern, Dalhousie University, 1334 Barrington Street, P.O. Box 15000, Halifax, NS B3H 4R2, Canada; 3MITACS Globalink Intern, Department of Mechanical Engineering, Coimbatore Institute of Technology, Coimbatore 14, Tamil Nadu 641 014, India

**Keywords:** phthalates, environmental contamination, extraction, detection

## Abstract

Phthalates (PAEs) are a group of synthetic esters of phthalic acid compounds mostly used as plasticizers in plastic materials but are widely applied in most industries and products. As plasticizers in plastic materials, they are not chemically bound to the polymeric matrix and easily leach out. Logically, PAEs should be prevalent in the environment, but their prevalence, transport, fate, and effects have been largely unknown until recently. This has been attributed, inter alia, to a lack of standardized analytical procedures for identifying them in complex matrices. Nevertheless, current advancements in analytical techniques facilitate the understanding of PAEs in the environment. It is now known that they can potentially impact ecological and human health adversely, leading to their categorization as endocrine-disrupting chemicals, carcinogenic, and liver- and kidney-failure-causing agents, which has landed them among contaminants of emerging concern (CECs). Thus, this review article reports and discusses the developments and advancements in PAEs’ standard analytical methods, facilitating their emergence from obscurity. It further explores the opportunities, challenges, and limits of their advancements.

## 1. Introduction

Phthalic acid esters (PAEs), popularly known as phthalates, are a class of lipophilic chemicals widely used as plasticizers and additives in plastics to improve plasticity, flexibility, durability, and resistance to UV degradation and combustion [1,2]. PAEs are used in nearly every industry and are broadly found in construction materials, printing inks, varnish, latex paint, cosmetics and personal care products, clothing, food packaging, pharmaceutics, medical products, intravenous cannulas, and insecticides. Some of the most widely used PAEs include butyl benzyl phthalate (BBP), dibutyl phthalate (DBP), di(2-ethylhexyl) phthalate (DEHP), di-isononyl phthalate (DINP), and di-isodecyl phthalate (DIDP).

Due to not being chemically bound to the materials they are added to, PAEs can be easily released from the products that contain them [3]. Consequently, the PAEs’ occurrence in the environment has been increasing drastically with their increasing use in recent years, and they have quickly gained prominence among environmental pollutants classified as contaminants of emerging concern (CECs). CECs are pollutants detected in water bodies that may cause ecological or human health impacts and are typically not regulated under current environmental laws [4]. Like most CECs, chronic exposure to PAEs can damage the liver, kidneys, and lungs. In addition, they can cause hormonal interference to reproductive hormones (e.g., luteinizing hormone, free testosterone, sex hormone-binding globulin) or thyroid function. Prenatal exposure to PAEs is associated with adverse impacts on neurodevelopment, including a lower IQ, problems with attention and hyperactivity, and poor social communication [5]. They can also bioaccumulate and persist in the environment, potentially building up in the food chain.

Until recently, despite being widely utilized and present in the environment, the prevalence, transport, and destiny of numerous PAEs were largely undiscovered, making them less noticeable on the radar of awareness regarding chemical pollution [6]. They have not gained as much prominence as microplastics despite having a higher potential for environmental hazards. Partially, this can be attributed to the challenges in determining PAEs in different environmental matrices that have mostly led PAEs to remain below the pollution radar. In addition, direct analysis of PAEs is complicated due to their low concentrations in complex matrices (Figure 1). However, the current advancement in analytical techniques, especially chromatographic quantification like tandem mass spectrometry, is paving the way for detecting PAEs at trace concentrations. In this regard, a few analytical procedures for the detection of PAEs have surfaced. Some are working well under specific conditions, while others still need optimization. Similarly, several sample preparation techniques, including solid-phase extraction [7,8], solid-phase microextraction [9,10], liquid–liquid extraction [11,12], microwave-assisted extraction [13,14], liquid-phase microextraction techniques [15], stir-bar sorptive extraction [16], and pressurized liquid extraction [17], among others, have been adopted for the detection of PAEs from different environmental matrices. Nonetheless, most procedures are not yet perfect but are improving daily.

In this review article, we explore the dynamic advancements in analytical methods and technology that have significantly contributed to uncovering the complexities of PAEs in the environment. With a primary focus on the period since the recognition of PAEs as contaminants of emerging concerns (CECs) in the early 2000s, our goal is to trace and highlight the progress made in sample handling, preparation, and detection techniques. This paper highlights the challenges, limitations, and opportunities throughout this journey while emphasizing the potential opportunities for further exploration. Additionally, this article provides a comprehensive overview of PAEs, shedding light on their physiochemical properties and toxic effects. This article is a product of an extensive preliminary literature review in the collaborative initiative between researchers in India and Canada to develop reliable, sensitive, and selective detection techniques for emerging phthalate contaminants in environmental matrices using tandem mass spectrometry. We systematically conducted comprehensive searches using various keywords, including phthalates, extraction, detection, and preconcentration, along with different analytical methods. Our exploration spanned multiple scientific databases, such as Google Scholar, PubMed, Medline, Scopus, Elsevier, etc. The identified articles were narrowed down by focusing on their relevance and recent publication dates. We selected articles published recently, prioritizing high-impact factor journals with full-text accessibility. The final compilation consisted of 100 eligible articles, categorized into two primary groups: the synthesis, impact, and toxicity of PAEs and the analytical procedures for the extraction and detection of PAEs.

## 2. Phthalates

### 2.1. Physicochemical Properties

PAEs are esters of phthalic acids. They are also known as esters of benzene-1,2-dicarboxylic acid. A benzene ring with an ester functional group is their structure character. They are commonly produced by a reaction of phthalic anhydride with alcohol. Their properties can be tuned by changing the alcohol type, allowing for an almost limitless range of products, although only around 30 are, or have been, commercially important. This reaction can be catalyzed via base catalysis or may happen at a reflux in hexane. Either of them concludes with the production of monoester PAEs. By having a high temperature or being catalyzed by metal complexes, heteropolyacids, or p-toluenesulfonic acid, this reaction produces diester phthalates (reactions are described in Figure 2). Recently, a report showed that the latter reaction can be performed at lower temperatures by having FeCl_3_ as the Lewis acid catalyst for esterification. This catalyst is a neutral and cheap material, enabling the production of PAEs commercially valuable [18].

PAEs, as oily liquids, have high boiling temperatures, good solubility in most organic solvents, and weak solubility in water. Their solubility in water decreases by extending their carbon chain or increasing their molecular weight. Due to their structure and physiochemical properties, PAEs have been good candidates for being plasticizers since 1921. Their high compatibility with various polymers enables them to disperse uniformly within polymer matrices, resulting in a homogenous, durable product. They have shown a considerably high plasticizer efficiency, ensuring that a small quantity of them can achieve the desired plasticizing effect, making them a cost-effective plasticizer. Their low volatility and high chemical stability help them remain within the plastic matrix longer, providing flexibility and softness for an extended time.

Furthermore, PAEs can help maintain the transparency and visual clarity of plastics, such as clear films, sheets, and packaging material. Although these advantages have kept these compounds useful in different industries, they do not bind permanently to the matrices where they are used and can migrate to their environment easily. This fact has raised lots of concerns about their fate and effects on the environment and human life.

### 2.2. Environmental Sources and Fates

PAEs are becoming ubiquitous environmental contaminants because of their widespread use, allied in almost all industries. They can enter the environment not only during the manufacturing process but also through the daily use of various produced items, such as food packaging, toys, paints, construction materials, personal care items, cosmetics (e.g., nail varnish), and electronic and medical devices (e.g., bags for intravenous fluids, breathing masks or umbilical catheters). Within any material, PAEs are not chemically bound to the matrices, allowing them to migrate to the surface of the products and accumulate in the environment by leaching, migration, and oxidation during manufacture, storage, usage, or disposal. For instance, the brittleness of some plastic materials over time or when exposed to sunlight is evidence of the release of phthalate into the environment [3].

PAEs can enter wastewater streams through domestic or industrial discharge. During the wastewater treatment process, they can accumulate in sewage sludge, which is commonly used as agricultural fertilizer. In addition to this pathway, soil can become contaminated with PAEs through leaks from agriculture machinery and the deposition of air or organic fertilizers. Once present in the soil, PAEs can migrate to the plants and other nutrients through the roots.

Due to their high molecular weight, PAEs have low vapour pressure and do not tend to evaporate into the air. Furthermore, they resist degradation and do not easily break down via natural processes, such as sunlight or microbial activities. Their hydrophobic nature causes them to bind to the organic compounds in the environment. These properties lead to their accumulation in the soil, water, and sediment for extended periods. Their resistant nature allows them to undergo long-range transport by wind, water currents, or other driving forces. This ability allows PAEs to spread beyond their original release point and impact remote areas. The combination of their high environmental resistance and ability to migrate over long distances results in a widespread distribution of PAEs in the environment. Once released, they can contaminate the air, soil, and water, entering the food chain through bioaccumulation (Figure 3).

### 2.3. Phthalates Effect

#### 2.3.1. Environmental Toxicity

Once PAEs reach water sources, they can affect aquatic organisms, ranging from fish to algae. The primary effect that has been extensively studied is their impact on reproductive systems. Certain PAEs can disrupt the endocrine system of aquatic organisms, especially fish, leading to impaired fertility and reduced successful hatching. They can also cause the feminization of male fish, alter the sex ratios of the fish population, and result in reduced fertility.

In addition to reproductive effects, PAEs can influence the growth, development, and survival rates of various aquatic organisms. Invertebrates are sensitive to PAEs and experience inhibited growth upon exposure. PAEs can also interfere with the photosynthetic process of algae, reducing their ability to capture sunlight, and convert it into energy. Since algae serve as the primary producer in the aquatic ecosystem, this disruption can trigger a chain reaction affecting higher trophic levels, leading to imbalances in aquatic food webs.

PAEs can reach plants through contaminated soil, fertilizers, water, or air. Plants cannot degrade or metabolize PAEs, so they translocate and accumulate in different parts of the plants. This accumulation can harm the plant and any organisms that consume it, leading to a broader distribution. Exposure to PAEs can reduce the germination rate of plants, resulting in alterations to the plant species’s population. They can also affect root elongation and branching, leading to stunted rooting. Since the roots are crucial for nutrient and water uptake, defective roots impact the overall health and productivity of the plants. PAEs in plant components can decrease chlorophyll content, reduce photosynthetic efficiency, and compromise plant growth and productivity. As mentioned earlier, PAEs are endocrine disruptors that affect the hormonal regulations of plants, such as auxins and cytokinins, consequently affecting their growth and development. These effects can alter flowering patterns and other aspects of plant productivity.

Animal species can be exposed to PAEs through contaminated water, plants, air, or food chains. These compounds adversely affect animal body tissues, including the liver, kidneys, and reproductive system. This effect is even more pronounced when exposure occurs during their developmental stages [19]. For instance, animals exposed to PAEs in their mother’s womb have shown a defective sperm concentration and activity, early puberty, and an increased risk of testicular cancer. In mammals, PAEs cause a decrease in sperm quality and abnormal testicular development. PAEs can bind to hormone receptors, leading to abnormal signaling and a wide range of physiological changes [20].

#### 2.3.2. Human Toxicity

The primary source of human exposure in the general population is ingesting food contaminated during production, processing, and packaging. However, indoor air exposure, cosmetic products, and contact with medical devices should be considered as the other possible sources of PAEs. PAEs have not only been detected in human urine, breast milk, and amniotic fluid but can also cross the placenta, leading to fetal exposure closely linked to maternal exposure [21].

Upon human exposure, PAEs undergo a metabolic pathway consisting of at least two steps. The first step is hydrolysis, followed by a conjugation process. In the hydrolysis step, which takes place in the intestine and parenchyma, diester phthalates are hydrolyzed by the catalytic activity of lipase and the catalytic activity of esterases, resulting in the formation of primary monoester phthalates. Although the first step in typical metabolism is often detoxification, the hydrolyzed structure regarding diester phthalates is more bioactive. Short-branched PAEs are primarily excreted in urine after the initial metabolic step as monoester phthalates. Long-branched PAEs undergo several biotransformations, such as hydroxylation and oxidation, before they can enter the second phase of metabolism and be excreted in the conjugated form in urine and feces. This conjugation process is usually catalyzed by the enzyme uridine 59-diphosphoglucuronyl transferase, forming a hydrophilic glucuronide conjugate that can be excreted in urine [22]. These metabolic pathway is summarized in Figure 4. Depending on their chemical structure, PAEs can have different half-lives in the body, ranging from hours to up to 2 days. After this time, a small portion of the phthalate may remain in the tissues. However, since PAEs are continually introduced into the environment and make their way into the human body, the duration and concentration of exposure can render them toxic.

Since PAE possesses endocrine-disrupting properties, high exposure concentrations can lead to a range of adverse effects, such as fetal death, cancer, malformations, liver and kidney injuries, and reproductive toxicity. Additionally, since PAEs can transfer from the mother to the fetus via the placenta and to neonates through breast milk, the potentially harmful effects during development are concerning. Neonates have lower levels of pancreatic lipase, which results in considerably reduced metabolic capacity compared to adults. In adults, 80–90% of urinary metabolites are conjugated with glucuronic acid, whereas in children, this pathway is not fully developed until three months old, indicating that this crucial clearance mechanism is not yet available for young infants. This lower clearance rate results in a higher internal dose of toxic metabolites. This may affect the development of the endocrine system, which plays a crucial role in various biological functions, such as sexual development and reproductive functions in adulthood [24].

The adverse effects of PAE exposure strongly correlate not only with the dose and timing but also with gender. Some studies have shown that these exposures may negatively impact males, leading to undesired changes in reproductive and developmental health in adulthood, such as altered hormone levels, reduced semen quality, decreased fertility, and increased risk of infertility. Although men can tolerate high concentration of each PAE before these effects become evident, there is evidence to suggest that a higher level of total PAEs can be found in infertile men compared to fertile ones. Additionally, a significant correlation has been found between the total amount of PAEs and normal sperm morphology, as well as the percentage of single-stranded DNA in the sperm.

## 3. Regulatory Framework

The toxicity of PAEs and their potential health effects have raised significant concerns, prompting both researchers and regulatory bodies to monitor the presence of different PAEs in various consumer products, such as food or cosmetic items, as well as the environment. By determining the level of PAEs in any sample, the assessment of their potential health risk, the identification of their source, and the pathway of their exposure can be studied.

Regulatory agencies, such as the Environmental Protection Agency (EPA) and the Food and Drug Administration (FDA), have established limits and guidelines for the permissible levels of PAEs in different matrices. These regulatory standards are vital to ensure consumers’ safety and protect human health. To meet these stringent requirements, developing accurate and precise analytical methods capable of detecting PAEs qualitatively and quantitatively has become a significant research focus in recent years.

Sensitive and reliable analytical methods are essential for accurately detecting even trace amounts of PAEs, as even low levels of PAE exposure can lead to adverse health effects. Achieving regulatory compliance requires advancing analytical techniques, including sample preparation and detection methods, which will be discussed in detail in the following sections.

## 4. Contemporary Analytical Procedure

### 4.1. Extraction or Pre-Treatment Methods: Potential and Challenges

In addition to sampling and homogenization, efficient preconcentration and cleanup steps are necessary for assessing PAEs due to the complexity of the matrices and their low concentration levels [25]. However, it is important to know that samples can easily become contaminated during laboratory activities by glassware, solvents, and reagents; analyzing PAEs in various matrices is a challenging task that requires precautions to avoid contamination.

A significant challenge in the analysis of PAEs is the proper preparation of blank samples for analytical measurements since they are widespread compounds present in the laboratory’s environment air [26], organic solvent, chemicals [27], as well as laboratory materials such as tubing, caps, and filter papers [26]. To minimize the risk of contamination, it is crucial to keep the PAE analysis as quick and simple as possible.

Depending on the sample matrices, which may contain lipids, various extraction methods are available. These include liquid–liquid (L–L) extractions or micro extractions with organic solvents, solid-phase extractions (SPE) using cartridges, and solid-phase micro-extraction (SPME). It is worth noting that when the matrix, such as milk or oil, contains lipids, phthalate analysis requires additional steps such as headspace SPME and gel-permeation chromatography (GPC). This is necessary because the aforementioned methods may co-extract lipids and organic compounds along with the PAEs.

In the following sections, we will review these methods, exploring their capabilities in extracting PAEs from different matrices and highlighting their potential limitation.

#### 4.1.1. Solid-Phase Extraction (SPE)

The brilliant properties of solid-phase extraction (SPE), such as ease of operation, semi-automation capabilities, low solvent consumption, high enrichment factor, and speed, have kept it in the spotlight as the dominant method for treating water samples for years [28]. Additionally, SPE can be fully automated via a direct connection to a chromatograph.

Various activated solid phases, shaped as discs or cartridges, have been employed to extract PAEs from water samples and were subsequently eluted with organic solvents. The efficiency of SPE relies on the selectivity and development of the sorbent. A wide range of polymeric sorbents with a hydrophilic–lipophilic balance, such as Oasis HLB and Strata X, have demonstrated their effectiveness in extracting different PAEs from aqueous samples using different organic solvents as eluents, including acetone, dichloromethane (DCM), ethyl acetate (EtOAc), methanol, and n-hexane. The choice of the sorbent-eluant influences the recovery rate of PAEs. In the case of water samples with low-suspended solid materials (SSM) (≤1 g/L), PAEs can be extracted through SPE without any prior filtration [29]. However, when dealing with higher levels of SSM, pre-filtration becomes essential, which may result in potential errors up to a 20% inclination of the total PAE concentration due to the contamination risks [30]. Commercial SPE cartridges are not the only option for use as the extractant in SPE. Different materials such as carbon nanotubes, florisil, Carbograph, bamboo charcoal, Nylon 6 nanofibers, and styrene-divinylbenzene have also been employed. While SPE offers advantages such as good reproducibility, high recovery values, and efficient sample throughput, it has limitations, including high backpressure, sorbent surface degradation, SPE column blockage, and occasional irreversible adsorption. In addition, it is worth noting that most commercial sorbents are only available in a plastic cartridge form, potentially leading to PAEs leaching into the samples and affecting the analysis results [31].

Dispersive solid-phase extraction (DSPE), in which the adsorbents are dispersed in the sample, offers different advantages. This includes applicability to a wide variety of PAEs, high cost-effectiveness, minimal use of glassware/plasticware, and easy automation, all of which help minimize potential PAEs contamination. Additionally, this procedure achieves a higher recovery rate than the traditional SPE without the concerns of cartridge or disk blockage or the need to control the sample flow rate [32]. Subsequently, magnetic sorbents were introduced to enhance PAEs’ preconcentration, leading to a more convenient, efficient, and faster extraction process known as magnetic solid-phase extraction (MSPE). Various materials, including ALG@C_18_-Fe_3_O_4_-TNs [33], Fe_3_O_4_@C18@Ba_2_þ-ALG [34], chitosan-coated Fe_3_O_4_-C_18_ MNPs [35], polythiophene-coated Fe_3_O_4_ [36], and polypyrrole-coated Fe_3_O_4_ [37], have been utilized for the extraction of PAEs from different samples. MSPE is considered a green technique that minimizes the environmental impact by having a high extraction efficiency, short analysis time, and convenient process, dealing with large samples. According to reports, when MSPE is combined with GC/MS, the limit of quantification for 16 PAEs can reach the range of 3.1–37 ng L^−1^ [38,39]. However, when using detection methods other than chromatography, co-extraction of other compounds may potentially interfere with the detection or quantification of PAEs, reducing reliability. This is one of the primary concerns associated with the MSPE.

#### 4.1.2. Solid-Phase Microextraction (SPME)

SPME is another method based on sportive extraction, where the analyte is extracted from the sample using a liquid polymer or a solid phase coated on a fiber. This method is considered environmentally friendly due to its nearly solvent-free approach and has gained popularity for PAE analysis. It has the potential to combine sampling, extraction, enrichment, and analysis into one single step, minimizing the risk of contamination for analyzing PAEs. After extraction, the SPME fiber can be transferred to the GC injection port for thermal desorption of PAEs, followed by analysis. A variety of polymers, including polydimethylsiloxane and divinylbenzene (PDMS-DVB), handmade polyaniline, and polyacrylate fibers, have been successfully used as solid coatings for SPME fibers for the analysis of the six main PAEs listed by the US-EPA [40]. However, since the capability of these polymers is limited for analyzing other PAEs, further investigation and improvement of this method are needed. While SPME can offer advantages and potentially overcome the column blockage issue compared to SPE, it also has its limitations, such as the cost of fiber coating and the possibility of PAEs cross-contamination when reusing the sorbent.

Stir-bar sorptive extraction (SBSE) is a sub-category of SPME in which the sorbent is coated on a stir bar and can be followed by thermal desorption. In SBSE, the sorbent amount is considerably larger, resulting in a higher sample capacity and lower LODs. Moreover, in the case of liquid samples, SBSE can eliminate the need for a cleanup step, potentially reducing the chance of PAE contamination. Despite these advantages, it is worth noting that SBSE requires a longer extraction time compared to SPME, limiting its suitability for daily use.

Molecularly imprinted polymers (MIP) can be used as a sorbent in DSPE, SPME, or SBSE, offering a predetermined selectivity over the extraction process. Using MIP for the analysis of PAEs can effectively eliminate the interferences from complex matrices more efficiently than traditional SPE. The advantages of using MIP as a sorbent include high selectivity and physical and chemical stability. However, the main limitations include the complex and time-consuming preparation process, low binding capacity, and poor site accessibility for the PAEs are the main limitations [41].

Another option in SPME is using a polymer monolith capillary as an extractant phase, known as polymer monolith microextraction (PMME). PMME offers advantages such as eliminating toxic solvents, higher mass transfer, and flexibility compared to traditional SPME [42]. However, it is worth noting that PMME may have a lower extraction efficiency for PAEs, which is its primary drawback [43].

#### 4.1.3. Liquid–Liquid Extraction (LLE)

LLE has been widely utilized for extracting PAEs from various samples due to its simplicity and convenience [44]. Since the transfer of the analytes between two liquid phases depends on their solubility in each phase, PAEs can be extracted from aqueous samples using suitable organic solvents. Common solvents for extracting PAEs include propanol and hexane, while adding an organic modifier like methanol can enhance the extraction of nonpolar PAEs such as DEHP and DNOP [45]. The efficiency of the extraction process can also be improved by adding inorganic salts and controlling pH. The separation of two phases can be facilitated through techniques like centrifugation, adding 20–150 g/L of NaCl, ultrasound, freezing, or vigorous stirring. Although this method has been one of the most popular extraction methods for PAEs, it has drawbacks such as being time-consuming, labor-intensive, requiring large volumes of toxic solvent, and the possibility of PAE contamination in the organic solvents, limiting its application.

To address the limitations mentioned above, various modifications have been made to LLE methods for the PAE extractions. One such modification is solid-supported LLE (SLE), in which the extractant phase is supported on an inert solid phase and packed into a disk or cartridge. After introducing the sample, the analytes were selectively eluted with a suitable organic solvent, minimizing interferences of the matrices [46]. However, overusing the extractant is limited due to its low stability.

Microporous membrane liquid–liquid extraction (MMLLE) is another modification of LLE, in which the analytes are extracted into an organic solvent separated from the aqueous sample by a hydrophobic porous membrane. This method is suitable for hydrophobic analytes and can be connected online to HPLC and GC. It has been used for different PAEs with an LOD level of ng mL^−1^ ranges [45]. The challenge with this method is selecting a suitable solvent that can extract both polar and nonpolar PAEs.

#### 4.1.4. Liquid-Phase Microextraction (LPME)

The miniaturized form of LLE, which limits the considerable volumes of organic solvent to a few microliters, is known as LPME. LPME offers several advantages, including simplicity, rapidity, low sample volume, low cost, and high enrichment factors. It can be categorized into three main types, each with potential sub-groups: (a) single-drop microextraction (SDME); (b) hollow-fiber liquid-phase microextraction (HF-LPME); and (c) dispersive liquid–liquid microextraction (DLLME) [47,48].

SDME, first reported in the 1990s, overcame many limitations of SPME and limited organic solvent consumption to a few microliters. It involves a micro syringe needle immersed directly into the aqueous sample (known as direct immersion or DI-SDME) or fixed above the sample (known as headspace SDME or HS-SDME). This method requires inexpensive equipment and combines extraction, preconcentration, and sample introduction into a single step, minimizing potential contamination. Although their method significantly reduced the amount of organic solvent used, using multiple solvents remained a drawback. Subsequent modification simplified these methods using single solvents like toluene or introducing innovations like the bubble-in-drop (BID-)HS-SDME method. With these improvements, the LOD of these methods reached an ng mL^−1^ level for up to 17 PAEs in aqueous samples [49,50]. The instability of the drop resulting in poor reproducibility and low sensitivity can be considered drawbacks of the SDME methods.

To address the drop instability issue, HF-LPME was introduced as a revolution in comparison to SDME. It uses a hydrophobic porous hollow fiber, such as polypropylene, connected on one side to the needle tip of a micro syringe while leaving the other end suspended in the sample solution to protect the single drop [51]. This method achieved LODs in the range of 0.23–0.69 μg L^−1^ for PAEs and offered advantages such as low cost, full automation capability, and the disposability of the used fiber, minimizing cross-contamination risks [49,50]. However, HF-LPME suffers from the unavoidable manipulation of hollow fiber when placing it at the needle tip, which can introduce contamination and lead to fluctuating results [52].

DLLME is a relatively new miniaturized technique that enhances the extraction efficiency due to its larger interface between the sample metric and the extractant. It was proposed in 2006 for PAE extraction due to its higher efficiency, simplicity, and rapidity. This method can be further improved by utilizing additional physical forces, resulting in techniques such as ultrasound-assisted DLLME (UA-DLLME), proposed for the extraction of six PAEs from milk [47], ultrasound-vortex-assisted DLLME (USVA-DLLME), established for the extraction of PAEs from wine [53], and magnetic stirring-assisted DLLME (MSADLLME), used for the extraction of PAEs from the aqueous sample [52]. Despite the significant advantages of DLLME, its application is limited by the availability of high-density solvents suitable for DLLME and compatible with PAEs. Additionally, the difficulty of automation and the common use of chloro-containing organic solvents, such as chlorobenzene, dichloromethane, chloroform, or carbon tetrachloride, which are considered environmental hazards, restrict its application. Ionic liquids, which are non-volatile and non-toxic, have been considered alternatives for PAE extraction [54]. However, their application is limited due to the high cost of preparation and low stability [55].

#### 4.1.5. Other Methods of Extraction

In addition to the previously mentioned methods, which have been the primary procedures for PAE extraction from various samples, other reported methods include the following:

*Cloud-point extraction (CPE)*—This procedure involves extracting PAEs into a very small volume of a non-volatile surfactant-rich phase. In the research by Ling et al., this method was coupled with UPLC, and the obtained hydrophobic analytes extract was analyzed to achieve LOD levels in the ng mL^−1^ range for PAEs [56]. CPE offers the advantages of using an environmentally friendly solvent and surfactant. However, since these surfactants are non-volatile, this method is not suitable for PAE detection via GC, which is designed for volatile compounds.

*Accelerated solvent extraction (ASE)*—This extraction method involves using an organic solvent at high temperature and pressure to extract analytes quickly and efficiently. ASE, when coupled with GC-MS, can be used for PAE extraction from various matrices, including food, soil, and plastics [57,58,59,60]. This method can be automated, offering benefits such as low solvent consumption, short extraction time, and high recovery. However, the high temperature and pressure during this process have the potential to hydrolyze PAEs, leading to the cleavage of carbon–oxygen bonds and isotopic enrichment of carbon and hydrogen, which can pose challenges for detection [61].

*Continuous-flow microextraction (CFM)*—In this method, the extraction solvent drop is injected into a glass chamber using a micro syringe and held at a tubing outlet where the samples flow through it and into the waste [62]. Continuous extraction occurs, resulting in a rapid process. The extraction solvent can then be transferred to GC-MS for analysis. When coupled with LC, this method can achieve LOD levels in the range of ng mL^−1^ for PAEs [63]. While CFM offers a continuous and low-labor procedure, maintaining the stability of the extraction drop during the dynamic process can be challenging.

### 4.2. Comparison between Different Methods of Extraction

The selection of an extraction method not only depends on the sample type and its physiochemical characteristics but can also be determined by the priorities and concerns of the analysis. Table 1 summarizes the advantages and disadvantages of the above-discussed methods from the perspective of PAE extraction.

### 4.3. Potential and Challenges of Contemporary Analytical Procedures

#### 4.3.1. Gas Chromatography (GC) Analysis

Due to the thermal stability and volatility of PAEs, GC coupled with a mass spectrometer is the most common detection method. This is generally conducted using a nonpolar column and Helium as the mobile phase [74]. PAEs can be analyzed in split or spitless mode, with or without pulsed mode. Additionally, it has been proposed that using a programmed temperature vaporizing (PTV) system can increase the amount of sample introduced into the column, resulting in better LODs of PAEs.

Using a mass analyzer as the detector or coupling it with other detectors, such as a flame ion detector (FID), for confirmation purposes makes the PAE detection method specific and sensitive with significantly low LODs and high resolution. Employing a combination of polar and nonpolar columns makes GC suitable for a wide range of PAEs, while utilizing a small diameter column helps to reduce the run time [75,76]. Despite these benefits, the high cost of the mass analyzer and the fact that it is a destructive technique limit its application [77]. Although there are some reports of using an electron capture detector (ECD) coupled with GC for PAEs analysis, unfortunately, this detector is only sensitive to halogenated compounds [78].

#### 4.3.2. Liquid Chromatography (LC) Analysis

LC can be a reliable method for analyzing PAEs due to its excellent selectivity. C18 is the commonly used column for PAE analysis because of the nonpolar nature of these compounds. Typically, a mixture of ACN/water and MeOH/water is the suitable mobile phase, often buffered or acidified by an isocratic or gradient elution to enhance the ionization efficiency [79,80,81,82]. Heating the column to temperatures between 25 and 80 °C can also improve separation.

Electrospray ionization (ESI) in positive mode is the most common interface for LC-based PAE detections. However, recent reports suggest that atmospheric pressure chemical ionization (APCI) can provide similar instrumental LODs and limit of quantifications (LOQs) for most PAEs [83].

LC can be combined with mass spectrometry (LC-MS/MS) for PAE detection, offering easier sample preparation and eliminating the need for derivatization. It can be a strong competitor to GC-MS, particularly for non-volatile PAEs. High-pressure liquid chromatography (HPLC) or ultra HPLC (UHPLC) are preferred for compounds that GC cannot adequately separate. Other detectors, such as diode-array detector (DAD) or UV spectroscopy, can be coupled with LC, providing LODs in the μg L^−1^ ranges for PAEs in various samples like wine or mineral water [54,84]. The main advantages of the LC analysis include its non-destructive nature and the ability to operate fully automated processes.

However, LC has limitations, including its high solvent consumption and possible co-elution of different analytes with identical chemical functionality [85]. Moreover, in samples containing multiple isomers of PAEs with varying toxicological importance, LC may not fully separate all the isomers, leading to inaccuracies in quantifying specific isomers. Additionally, LC is a time-consuming technique, especially for complex samples requiring long-programmed elution after extensive sample preparation. This can limit the throughput and scalability of PAE analysis in daily experiments.

#### 4.3.3. Micellar Electrokinetic Capillary Chromatography (MEKC)

MEKC is a capillary electrophoresis method that combines electrokinetic chromatography and micellar chromatography, allowing the separation of PAEs with several advantages, including high efficiency, high output, and less consumption of reagents. In this method, analytes are separated by different partitioning between micelles and the surrounding aqueous buffer solution. MEKC has been employed to analyze PAEs in various sample matrices such as water and soil. It has been coupled with DLLME and diode-array detection to achieve a limit of quantification (LOQ) of 2.7 µg/L for PAEs [86].

The main limitation of this method lies in the limited solubility of certain PAEs in the micellar phase. Additionally, potential interferences from background substances can pose challenges. Furthermore, optimizing the surfactant concentration, pH, and ionic strength of the mobile phase can be time-consuming and may be considered a limiting factor for the speed of analysis.

#### 4.3.4. Fourier Transform Infrared Spectroscopy (FTIR)

FTIR, which provides a fingerprint signal for any analyte and offers high output, speed, simplicity, high sensitivity, and internal calibration, can be employed to detect PAEs in various physical states of samples. Although FTIR is more cost-effective, rapid, and involves fewer essential sample preparation steps than GC-MS, it is less sensitive and can detect the total amount of PAEs rather than specific phthalate compounds. Therefore, it is better suited for pre-screening, often coupled with GC-MS or other separation techniques, to achieve accurate detection and quantifications [87].

#### 4.3.5. Colorimetric Analysis

Colorimetric methods, which involve measuring an analyte with the aid of a color agent, using organic or inorganic compounds, and with or without an enzymatic step, have been widely utilized to determine PAEs. For instance, anhydrous phthalate can be hydrolyzed with sodium hydroxide and then dehydrated to form phthalate anhydride. Subsequent reaction with resorcinol in concentrated sulfuric acid results in the conversion to fluorescein, forming a color. This color change can be detected using absorbance spectroscopy with LODs of less than a μmol, depending on the phthalate type [88].

Diffused reflectance UV spectrometry, coupled with membrane filtration, has also been employed for PAE detection with LODs in the range of a few μg/L and recovery rates between 99% and 105%. In this technique, the analyte passes through the membrane and can be quantified directly using a UV spectrophotometer equipped with an integrating sphere. The accuracy and sensitivity of this method can be further enhanced by utilizing nanoparticles. Gold nanoparticles (AuNP) are commonly used for colorimetric detection due to their unique plasmonic effect. Recently, DNA-modified gold nanoparticles have been reported to enhance the selectivity and simplicity of PAE detection. PAEs induce nanoparticle aggregation, leading to a color change that can be detected using a UV-vis spectrophotometer. The LOD for this method has been reported to be less than 1 ppm, depending on the type of PAEs and the nanoparticles used [89].

In other reported methods, a crosslinker such as Cu^2+^ can be used alongside modified nanoparticles. Colorimetric methods offer several key advantages, including simplicity, cost-effectiveness, sensitivity, and speed. However, their primary limitation lies in their lack of sensitivity when dealing with complex samples. The possibility of interference from sample matrices necessitates sample preparation and manipulation. The sample preparation step can be time-consuming and a potential source of PAE contamination, resulting in fluctuated data.

#### 4.3.6. Immunoassay-Based Techniques

Immunoassay sensors that utilize immunochemical reactions coupled with a transducer provide specific recognition for detecting PAEs, particularly from plastic matrices. Several reported methods for quantifying PAEs in water samples are fluorescence-based, offering LODs in ng L^−1^ ranges and high recoveries [90,91]. Besides the limited required sample preparation, the main advantages of this method include its reliability and selectivity, which allows the analysis of the samples without any specific manipulation. This property ensures accurate PAE quantification due to the minimal contamination risk during sample preparation.

However, it is important to note that because these sensors rely on immunochemical reactions, they may be limitations when analyzing certain PAEs, necessitating additional sample preparation or processing steps to enhance the accuracy. Furthermore, immunoassay sensors are relatively expensive with a limited shelf life, which makes them less practical for continuous PAE analysis.

## 5. Emerging Trends and New Perspectives

In the recent decade, considerable advancements have been made in phthalate extraction and analysis techniques. Numerous pre-treatment techniques and detection methods can now be applied to identify and quantify these compounds in various environmental matrices, including atmospheric aerosols, indoor and outdoor air, municipal solid waste compost, sludge, fresh water and marine waters, soil, and sediments. Consequently, our understanding of the fate and effects of PAEs is improving daily. For instance, the potential risks to human health and ecosystems were evaluated more accurately than a decade ago. Further, predictions of their pollution trends and assessment of remediation effects are becoming more accurate. Sample treatment methods for PAEs in different matrices are evolving, with the most commonly used methods including liquid–liquid extraction, liquid–liquid microextraction, solid-phase extraction, solid-phase microextraction, and their derivative techniques. Green extraction protocols have gained prominence, leveraging environmentally friendly solvents and innovative sample preparation techniques [92]. For instance, the utilization of deep eutectic solvents in extraction processes has demonstrated both high efficiency and environmental sustainability, offering an alternative to traditional organic solvents [93].

The primary separation and detection methods are GC with flame ionization or mass spectrometry and HPLC with ultraviolet or mass spectrometric detection. These analytical techniques have witnessed a significant shift towards high-throughput screening methods, aiming for rapid and comprehensive assessments of PAEs. Liquid chromatography–mass spectrometry (LC-MS) and gas chromatography–mass spectrometry (GC-MS) techniques have been refined and optimized for improved sensitivity and specificity. For instance, the isotope dilution method has been introduced as a powerful technique for enhancing the accuracy and precision of PAE quantification. By accurately quantifying the target analytes, isotope dilution contributes to the robustness of the analytical procedure, particularly in complex environmental matrices [94,95,96]. Ultra-high-performance liquid chromatography (UHPLC) coupled with high-resolution mass spectrometry (HRMS) has shown promise in achieving enhanced separation and detection capabilities as well [97].

Machine learning algorithms and data-driven approaches have become integral in handling the complexity of phthalate datasets [98,99]. These computational methods allow for the extraction of meaningful patterns, aiding in the interpretation of intricate relationships within large datasets. This synergy between analytical chemistry and data science holds the potential to revolutionize the predictive modeling of phthalate exposure and associated risks.

Furthermore, research efforts have delved into the development of novel adsorption materials for the extraction of trace phthalates from different matrices. Different novel materials such as molecularly imprinted polymers (MIPs) [100], microporous polymers [9], and metal–organic frameworks [101] have shown promising potential as absorbents to extract PAEs from various sample matrices.

Despite these advancements in the PAE detection pathway, the challenges in analyzing phthalate in environmental samples still exists, making a call for further enhancements. These challenges originate from two perspectives. First, the real concentrations of phthalate contaminants are often underestimated in environmental samples because they exist in complex matrices where contaminants are typically present in trace levels. These trace concentrations of PAEs in environmental samples make the development of efficient pre-treatment methods for extracting the target contaminants extremely challenging. Additionally, these trace levels can suppress the signal intensity of the analytical instrument at the interface of the system when several matrix components co-elute. As a result, there is a need to develop sensitive and robust methods for extracting and determining a wide variety of PAEs. For instance, spiking methods and careful sample reconstitution are highly recommended [102]. The second challenge arises from the ubiquity of PAEs in everyday human life. They are pervasive in all environmental compartments, including laboratory environments and products contained in materials, reagents, and equipment. Thus, the risk of sample contamination during the analytical procedure due to the ubiquity of PAEs in the laboratory environment, products, solvents, and reagents may often lead to false positives or overestimated results.

In the future, there will be a need to develop novel adsorption materials for extracting trace PAEs from different matrices. It is hoped that this advancement will take advantage of nanotechnology and the known versatility of nanomaterials to develop even more effective nano-based absorbents. Similarly, it is also necessary to develop new analytical techniques capable of online or on-site detection, such as optical and colorimetric sensors and molecular chemistry systems like micro-total analysis systems, surface-enhanced Raman scattering systems, and surface plasmon resonance systems.

While these advancements offer promising avenues for enhanced PAE detection, there are inherent challenges and limitations that must be acknowledged. Firstly, the development of novel adsorption materials utilizing nanotechnology may face hurdles related to the scalability and cost-effectiveness of these materials. The transition from laboratory-scale experiments to large-scale environmental applications might encounter practical obstacles. Additionally, the environmental impact of manufacturing and disposing nanomaterials warrants careful consideration, aligning with sustainable practices. Secondly, the deployment of new analytical techniques for online or on-site detection, such as optical and colorimetric sensors, may encounter challenges in terms of sensitivity and selectivity. The real-world application of these techniques requires meticulous validation and calibration to ensure reliable results across diverse environmental matrices. Moreover, the integration of these advanced technologies into routine monitoring systems demands careful consideration of operational complexities and the need for skilled personnel. In navigating these limitations, future research should focus on addressing these practical challenges to ensure the successful translation of these advancements from the laboratory to real-world environmental monitoring scenarios.

## 6. Concluding Remarks

In conclusion, PAE detection has witnessed significant progress in recent years, with promising insights into extracting and quantifying these compounds in various environmental matrices. While advanced pre-treatment techniques and detection methods have facilitated a better understanding of the presence, fate, and potential effects of PAEs, challenges in phthalate analysis remain insignificant.

The primary obstacle lies in comprehending the true concentrations of PAEs, given their trace levels within complex matrices. This necessitates the continual development of efficient pre-treatment methods and highly sensitive analytical techniques. Additionally, the omnipresence of PAEs in laboratory environments, materials, and chemicals poses an ongoing risk of sample contamination during analysis, potentially leading to false positives or overestimated results.

Looking to the future, continued innovation will be crucial. Novel adsorption materials, especially ones leveraging nanotechnology, hold the potential for the effective extraction of PAEs from diverse matrices. Furthermore, developing advanced analytical techniques, such as optical and colorimetric sensors and surface-enhanced Raman scattering, promises greater accuracy in the qualification and quantification of PAEs.

In summary, the journey of PAE extraction and analysis has been marked by significant progress, reflecting the collective efforts of researchers committed to understanding the impact of these compounds. Despite the challenges, the expanding knowledge and evolving technologies are bringing us closer to establishing comprehensive strategies for the management PAEs in the environment and safeguarding humans and ecosystems from the potential threats posed by these compounds.

## Figures and Tables

**Figure 1 molecules-29-00106-f001:**
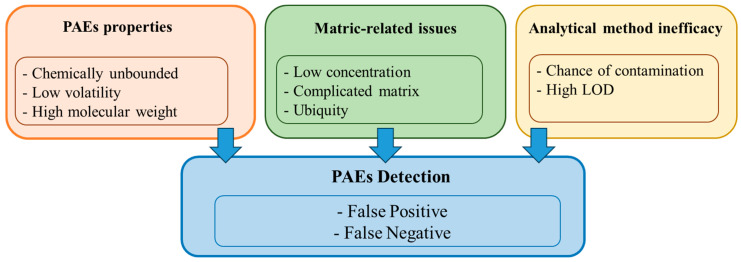
Factors contributing to PAEs’ detection uncertainty.

**Figure 2 molecules-29-00106-f002:**
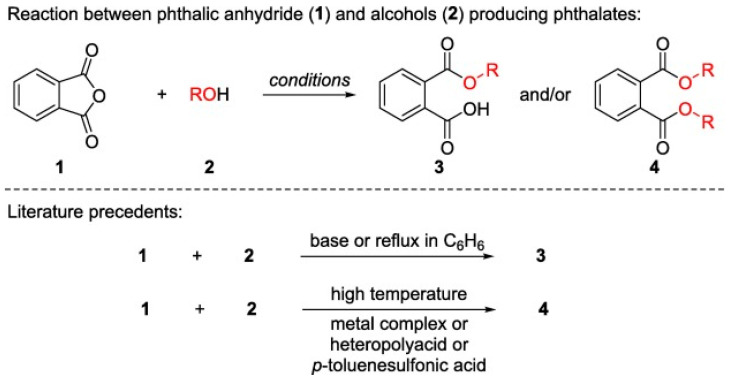
Synthesis of PAEs using base- or metal compound-based catalysis [18].

**Figure 3 molecules-29-00106-f003:**
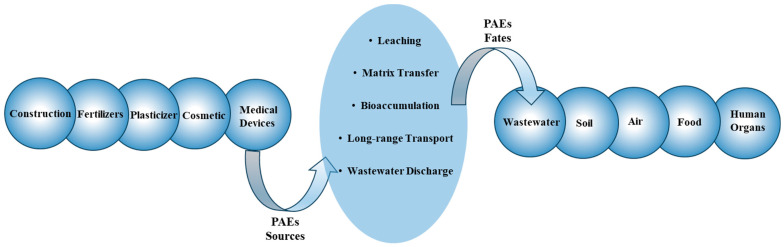
Sources and fates of PAEs in the environment.

**Figure 4 molecules-29-00106-f004:**
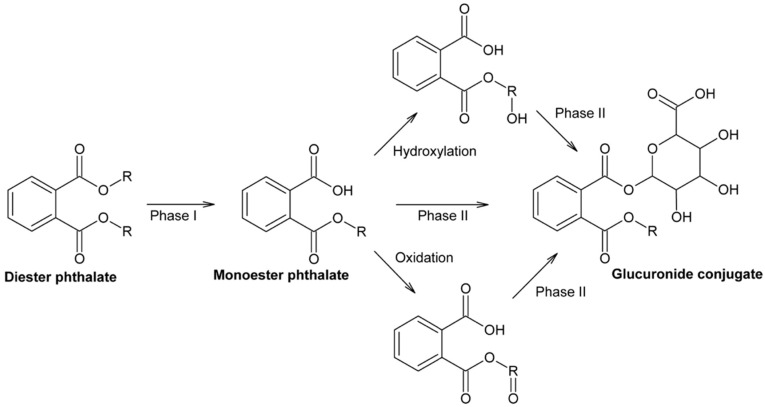
Metabolic pathway of PAEs [23].

**Table 1 molecules-29-00106-t001:** Comparison of different extraction methods.

Method	Advantages	Disadvantages
SPE	Simplicity, accuracy, high throughput and high recovery, low solvent consumption, ease of automation	Inability to extract from large sample volumes, susceptibility to sorbent vulnerability, high probability of column blockage [64]
SPME	Simplicity, rapidity, minimal solvent usage, ease of automation	The short lifespan of the fiber, high cost, potential for cross-contamination [65]
SBSE	High sample capacity, high recovery and sensitivity, and low detection limits eliminate the need for a cleanup step in liquid samples.	Limited repeatability [66]
LLE	Simplicity, convenience, popularity	Time consuming, labor intensive, requires large sample volumes, involves toxic organic solvents, and is inapplicable for trace analytes [67]
LPME	Low cost, limited organic solvent consumption, simplicity and possibility of full automation, low chance of cross-contamination	Time consuming, limited sample volume [68]
SLE	Limited organic solvent consumption, higher selectivity compared to LLE extraction	Potential of cross-contamination, low stability of the extractant [69]
MMLLE	Capability to operate online with GC and HPLC	Limited selection of organic solvents suitable for all PAEs [45]
SDME	Limited organic solvent consumption, fast merging sample preparation, preconcentration, and introduction step minimized the risk of cross-contamination.	Requires multiple solvents [49]
HF-LPME	Capability for full automation, minimization of cross-contamination	There is a high risk of contamination during the fiber placement process [70]
DLLME	Simplicity, high efficiency, rapidity, low sample volume requirement, cost-effectiveness, high enrichment factor	Use of toxic organic solvents, difficulty of automation, high-cost preparation process, low stability of the extractant drop [71]
CPE	Environmentally friendly	Incompatibility with GC [72]
ASE	Compatibility with different matrices, fast and low-solvent consumption	Utilizes harsh physical conditions and has a high risk of detection errors [57,73]
CFM	Rapid and online extraction	Limited reproducibility [63]

## Data Availability

No new data were created or analyzed in this study. Data sharing is not applicable to this article.

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
