# Peer review of "Unpacking Phthalates from Obscurity in the Environment"

_molecules, 2023, doi:10.3390/molecules29010106_

Round 1
Reviewer 1 Report
Comments and Suggestions for Authors
The article is informative and can be accepted after correction, few of which are suggested below:
1. Grammar of the text needs improvement.
2. Font size should be consistent.
3. Challenges of the advancements are described but limits of the advancements should also be discussed as the author mentioned in abstract.
4. It would be interesting if there is emphasis on recent research on phthalates and emerging trends to keep the information up-to-date.
5. Visual elements like graphs, tables, and figures should be included as they provide better understanding of the data.
6. Statistical analysis is not mentioned.
Question # 1
Authors has synthesized the esters of phthalic acids and claims that this synthesis may happen at the reflux in hexane (in synthesis it is inappropriate to say “may happen at the reflux”)
The author should give some proof of this statement. Or some reference to prove it?
Question # 2
This reaction is a simple reaction between phthalic anhydrides and alcohol to produce the esters. In this case alcohol react in two ways, first, it generates the -OR group for ester and secondly, it acts as solvent for reflux. Then why there was a need to mention the solvents “Hexane” for reflux?
And why there was a need to add the metal complexes, hetero polyacids, or p-toluene sulfonic acid. Give some specific reason for this?
Question # 3
The conditions for the reactions are not mentioned elsewhere in the manuscript. Please provide the proper procedure for this reaction.
Question # 4
If the author claims that he synthesizes the ester ofphthalic anhydride. Please provide proof of final product with some characterization techniques like FT-IR, Mass, 1H-MNR and 13C-NMR as they are mandatory acceptance of the manuscript.
Comments on the Quality of English Language
1. Grammar of the text needs improvement.
2. Font size should be consistent.
Reviewer 2 Report
Comments and Suggestions for Authors
This manuscript is a good contribution as a review of analytical methods for identification/determination of phthalates in complex matrices. Both sample treatment and analyses are dealt with appropriately. Overall phthalate levels and speciation are considered. The paper is generally well-organized and would be a good addition to the journal.
I do have several edits/suggestions that ought to be examined before publication. They should be able to be disposed of in short order.
Minor adjustments:
Line 51, reword a bit something along the line of ‘While widespread and ubiquitous in the environment,….’
Line 60, not sure what reciprocally means in this context, I’d recommend deleting it
Line 307, nearly rather than neerly
More substantive adjustments
Paragraph at line 495 could use a rewrite. IT reads as if FTIR and ICR-MS are a tandem/hybrid process which I don’t feel to be the case. Also, there does not appear to be a reference for the ICR work, such as in
https://doi.org/10.1002/cjoc.201400564
Finally, given the difficulties mentioned in quantitation, I would recommend some reference to the use of isotope dilution experiments, such as in https://doi.org/10.1016/j.jchromb.2004.10.056
Such reference could come in a choice of several locations, either the LC section (which is mostly LC-MS) or in new perspectives
Once attention is paid to these points I think the manuscript is ready for publication
Comments on the Quality of English LanguageAll points dealt with in the single file
Reviewer 3 Report
Comments and Suggestions for Authors
This review manuscript by Marzieh Baneshi et al. (T Unpacking Phthalates from Obscurity in the Environment) deals with a typical chemical-phthalates for its physicochemical properties, environmental effects, analytical procedure and so on. It is no doubt that this topic is very interesting and important to explore the potential ecological and human risk. And it seems to own the potential application values.
However, it must be mentioned these issues, including but not limited as follows,
1, note, ‘Phthalates’ occurs at some part, ‘PAE’ is also used for some section. Please keep in step.
2, what is the aim of section 2 and 3, still as background information?
3, Introduction part, please make research purpose more clearly.
4, Overall, this manuscript is mostly a subjective description. Please give more objective evidence with publications.
5, It seems that Table 1 is very important for this draft. Please show the origins, sample information and more details in this Table.
6, please give more English Check.
In summary, this recent version is not up to the level of this journal.
Comments on the Quality of English Languageplease make more English check.
Round 2
Reviewer 3 Report
Comments and Suggestions for Authors
Thank you so much for your revision.
Herein, I have no more comment.
Please follow the decision from editor.
Comments on the Quality of English LanguageMinor editing of English language required